# Assessing the Effects of Refuse-Derived Fuel (RDF) Incorporation on the Extrusion and Drying Behavior of Brick Mixtures

Ioannis Makrygiannis [1,*], Athena Tsetsekou [1], Orestis Papastratis [1] and Konstantinos Karalis [2]

1 School of Mining Engineering and Metallurgy, National Technical University of Athens, Zografou Campus, 15780 Athens, Greece; athtse@metal.ntua.gr (A.T.); orestep2018@gmail.com (O.P.)
2 Institute of Geological Sciences, University of Bern, CH-3012 Bern, Switzerland; konstantinos.karalis@geo.unibe.ch
* Correspondence: ymakrigiannis@sabo.gr

**Abstract:** This study explores the potential benefits of incorporating Recycled Demolition Waste (RDF) as an additive in ceramic mass for the brick industry, with a focus on applications such as thermoblocks. The research underscores the significance of sustainable waste management practices and environmental conservation by diverting waste from landfills. RDF, exhibiting combustion properties above 550 °C, emerges as a valuable candidate for enhancing clay-based materials, particularly in the brick production process where firing temperatures exceed 850 °C. Conducted in two phases, the research initially concentrated on RDF preparation, RDF integration with clay materials, and its influence on extrusion and drying phases. Employing innovative techniques involving brick and tile industry machinery coupled with sand incorporation yielded promising results. The grounding of RDF particles to less than 1 mm not only facilitated the mixing process but also ensured stable grinding temperatures within the hammer mill, reducing operational costs. During extrusion, challenges associated with unprocessed RDF material were addressed by utilizing ground RDF, leading to a more efficient and cost-effective process with enhanced plasticity and reduced water requirements. Practical implications for brick plant operations were identified, promoting resource and energy savings. Drying behavior analysis revealed the positive impact of RDF integration, showcasing reduced sensitivity, decreased drying linear shrinkage, and improved density properties. RDF's role as an inert additive resulted in a 5% reduction in density, enhancing porosity and thermal insulation properties, particularly in thermoblock applications. In the brick industry, where durability, thermal performance, and cost-efficiency are paramount, this study emphasizes the potential benefits of incorporating RDF into clay-based materials. While further research is needed to address the firing procedure of RDF as a brick mass additive, the initial findings underscore the promise of this approach for sustainable and environmentally responsible brick production. This study contributes to the literature by shedding light on the advantages and challenges of integrating RDF into clay-based products, supporting sustainability and waste reduction in construction and manufacturing. The findings provide valuable insights into the performance and feasibility of these mixtures, offering crucial information for industries striving to adopt eco-conscious production methods. This article not only outlines the applied methodology and experimental setup but also presents results related to the behavior of RDF-inclusive clay block mixtures in the production environment. Anticipated to exert considerable influence on future practices and policies, this research contributes to the growing body of knowledge concerning eco-friendly and sustainable manufacturing processes.

**Keywords:** brick production; sustainable materials; material incorporation

## 1. Introduction

Bricks, one of the oldest known building materials, have long stood the test of time, demonstrating remarkable resistance to the most challenging weather conditions. This inherent resilience has made them the cornerstone of construction, heralding bricks as

the most dependable building material in the industry. Bricks come in various types and possess distinctive properties rooted in their size, composition, weight, and shape. Ceramic bricks, in particular, exhibit a set of key characteristics including strength, durability, dimensional stability, longevity, and fire and weather resistance. These attributes render them versatile and indispensable in numerous aspects of building construction, from cladding, partitions, and structural walls to pavements and chimneys. Globally, an astounding 1600 billion bricks are produced annually, underscoring their paramount role in the construction landscape [1].

The manufacturing process of ceramic products shares a commonality, regardless of the materials employed or the intended final product. This process unfolds through several well-defined stages: the extraction of raw materials, the transportation and storage of these materials, the preparation of raw materials, shaping, drying, and, ultimately, firing [2].

This work aims to showcase that the integration of Refuse-Derived Fuel (RDF) into the ceramic mass of clay blocks can effectively enhance their thermal insulation characteristics by decreasing their final body density [3]. The primary focus is on assessing the impact of this incorporation on the production process and the final product, with a particular emphasis on the extrusion and drying phases. According to EN1745 (Annex A), a clay's density affects the coefficient of thermal insulation ($\lambda_{10,dry,mat}$). A lower density leads to lower $\lambda_{10,dry,mat}$ values, which gives better thermal insulation properties. By enhancing the thermal insulation properties of clay blocks, we anticipate a reduction in heat loss, which, in turn, offers improved temperature maintenance. This improvement has the potential to yield energy savings and financial benefits. Bricks are a fundamental element in construction, and enhancing their thermal insulation properties carries substantial economic and environmental implications, particularly when utilizing RDF as a recycled waste material to achieve this enhancement.

Furthermore, integrating RDF into the ceramic body mass on an industrial scale proves feasible without exorbitant expenses or substantial modifications to existing production environments. RDF is a lightweight material and can be seamlessly introduced into the production line via a dedicated system [4]. The transportation of RDF from production facilities to brick manufacturing sites poses no logistical challenge as RDF's inherent lightness and compressibility allow for its cost-effective and secure transit via various means of transportation.

In an era characterized by sustainable practices and heightened environmental awareness, this study embarks on an exploration of innovative approaches within the realm of construction and manufacturing. Our investigation centered on the behavior of three distinctive brick mixtures throughout the production process, encompassing pre-crushing, extrusion-forming, and drying stages. The first mixture, consisting of unadulterated clay, served as a control for comparative analysis. The second mixture blended non-hazardous RDF with clay in a 10% volume ratio, while the third introduced a pioneering element—10% milled RDF content. The leaching process of RDF was executed through a hammer mill, intimately mixing RDF with silica sand. Our overarching objective was to comprehensively unravel the behavior of the wet and dry brick products at each production stage [5].

In summary, this study endeavors to address fundamental questions concerning the behavior and viability of RDF-enhanced clay block mixtures in the production realm. By doing so, it actively participates in the ongoing endeavor to minimize environmental impacts within the construction and manufacturing sectors [6].

The primary objective of this study was to delve into the intricate relationship between the grain size of Refuse-Derived Fuel (RDF) and its impact on various pivotal aspects of the brick manufacturing process. Specifically, the research aimed to elucidate how RDF grain size influences the molding of wet bricks during vacuum extrusion and the precision of cutting the extruded wet bodies into standardized dimensions [7]. Furthermore, the study sought to assess the sensitivity of the drying process within RDF-infused mixtures and provide a comprehensive comparative analysis of drying-related metrics, including weight

losses, linear drying shrinkage, densities of the final dried samples, bending strength, and the reabsorption of moisture from the surrounding atmosphere. The results of these meticulously conducted experiments offer profound insights, shaping our understanding of the feasibility and implications of integrating RDF into ceramic mass, particularly concerning the development of environmentally friendly "green bricks".

In the exploration of sustainable practices within the realm of ceramic material development, this study delves into the intricate interplay between Recycled Demolition Waste (RDF) and traditional clay mixtures. The extrusion process, a pivotal stage in ceramic production, unfolds as a dynamic arena where the influence of processed and unprocessed RDF becomes palpable. Notably, the processed RDF infusion introduces an intriguing shift in plasticity, necessitating additional extrusion water. In contrast, unprocessed RDF presents a paradox, showcasing improved plasticity while concurrently posing challenges during the subsequent cutting phase due to the size of RDF particles. This sets the stage for a nuanced investigation into the mechanical and drying characteristics of the resulting ceramic products.

The drying phase unravels a multifaceted narrative, unraveling insights into the sensitivity, shrinkage, and mechanical strength of clay blocks infused with RDF. The incorporation of RDF, irrespective of its processing state, emerges as a mitigating factor for sensitivity during the drying process. Yet, the reduction in bending strength, a pivotal mechanical attribute, becomes apparent, underscoring the profound impact of RDF on the material's microstructure. Porosity takes center stage in understanding this shift as RDF introduction leads to larger pores, compromising the structural integrity of the clay blocks. Furthermore, the consequences extend to density reduction, presenting a trade-off between weight considerations and the structural robustness of the final product. As this study unfolds, it navigates the intricate dynamics between RDF incorporation, porosity, and mechanical strength, shedding light on the delicate balance required in sustainable ceramic material development.

The investigation deepens its inquiry into the post-drying behavior of the ceramic products by scrutinizing their re-absorption characteristics. The observed increase in re-absorption percentages following the drying phase unveils the porous nature induced by RDF, emphasizing the material's responsiveness to environmental conditions. These findings illuminate the challenges and opportunities presented by RDF in construction applications, where exposure to varying environmental conditions is an inherent reality. As the study advances, it not only contributes to our understanding of the complex relationship between RDF, porosity, and moisture uptake but also underscores the need for a holistic comprehension of how these variables collectively shape the long-term performance and durability of ceramic materials in real-world scenarios.

## 2. Materials and Methods

### 2.1. Materials

The prepared mixtures underwent a systematic labeling process, as outlined in Table 1. Subsequently, a comprehensive series of tests were conducted to evaluate their performance across various stages, encompassing preparation, extrusion, shaping, and drying properties. The results of these tests are meticulously documented in Table 2, providing a detailed insight into the characteristics and behaviors of the constructed mixtures throughout the production process.

**Table 1.** Mixtures constructed during study.

| | Clay Material | RDF as Obtained | RDF (Milled) |
|---|---|---|---|
| | wt.-% | wt.-% | wt.-% |
| TZ | 100 | - | - |
| TZRDF10 | 90 | 10 | - |
| TZRDF10P | 90 | - | 10 |

**Table 2.** Qualitative and technological tests that took place in this study.

| Qualitative Tests | Technological Tests |
|---|---|
| Calcium carbonate | Plasticity/necessary water for extrusion |
| Grain size | Extruding |
| | Drying sensitivity |
| | Drying results |

Three distinct mixtures were formulated and evaluated in this study. The first mixture, designated as "TZ", consisted solely of plain clay material and the necessary water for extrusion with a plasticity index of 0.77 Pfefferorn, which was calculated as 17.44%. The second mixture, labeled "TZRDF10", combined clay material with 10% raw RDF and was constructed with 18.71% necessary water for a plasticity index of 0.77 Pfefferkorn, while the third mixture, labeled as "TZRDF10P", consisted of clay with 10% processed RDF. The necessary water for extrusion on this mixture was measured as 19.05% for a plasticity index of 0.73 Pfefferkorn. In all mixtures, the requisite amount of water was added to facilitate uniform blending. It is noteworthy that the clay material utilized in all three cases originated from the same quarry to isolate the impact of RDF exclusively. The clay material, sourced from CHALKIS S.A. in the Vasiliko Evia region, has been employed for tile production at the factory for over two decades and falls under the classification of inorganic clay with moderate plasticity, according to ISO 14688-2:2017 [8].

The chemical composition of the clay was determined through analysis that was conducted using Atomic Absorption Spectrometry (AAS), adhering to ISO 26845:2016 [9] standards. Furthermore, the granular characteristics of the TZ material were examined in accordance with ASTM D422-63 (2007) [10]. The clay's average density, assessed following ASTM D698-12 [11], stood at approximately 1781 Kg/m$^3$. Upon transportation from the manufacturing site to the laboratory, the clay exhibited an average moisture content of 8.15% [12]. In accordance with ISO 14688-2:2017 classification, TZ can be described as an inorganic clay with a medium level of plasticity.

Refuse-Derived Fuel (RDF) stands as a renewable energy source derived from Municipal Solid Waste (MSW), processed through mechanical and thermal treatments involving shredding and dehydrating waste materials, with the subsequent removal of non-combustible components [13]. The resulting RDF, comprising organic and combustible elements, proves advantageous as a supplementary additive in brick production [14]. Procured from a prominent waste management company in the Mediterranean, the RDF used in this study underwent differentiation between raw and processed forms due to their distinct impacts on the production process. Laboratory procedures included quality control focusing on material properties and the extrusion process, alongside technological control involving tests simulating factory conditions [15]. Comprehensive testing covered various aspects such as calcimetry, granulometry, Pfefferkorn plasticity, specimen manufacturing, drying cycle evaluation, bending strength determination, re-absorption capacity assessment, and density calculation. This meticulous approach facilitated a thorough examination of how RDF incorporation influences each stage of the brick manufacturing process and its potential for enhancing the production environment in a brick plant. In parallel, the selection of appropriate RDF as an additive in ceramic mass for brick production necessitates a careful evaluation based on key criteria, including calorific value, particle size, chemical composition, combustion characteristics, cost-effectiveness, environmental impact, regulatory compliance, and compatibility with production machinery. Manufacturers can leverage these criteria to make informed decisions that align with sustainability and efficiency goals in brick production processes. The physical properties, metal content halides, and sulfur and organics can be seen in the following Tables 3–6.

**Table 3.** Physical properties of RDF.

| Parameter | Unit | LOQ | Result |
|---|---|---|---|
| Ash 550 °C | Mass % d | 0.1 | 16.1 |
| Ash 815 °C | Mass % d | 0.1 | 15.2 |
| Moisture | Mass % ar | 0.1 | 9.2 |
| Volatile matter | Mass % ar | 0.1 | 58.7 |
| Volatile matter | Mass % d | 0.1 | 64.6 |
| Fixed carbon | Mass % d | 0.1 | 20.2 |
| Dry mass (105 °C) | Mass % ar | 0.1 | 90.8 |
| Carbon | Mass % d | 0.1 | 50.0 |
| Hydrogen | Mass % d | 0.1 | 5.76 |
| Nitrogen | Mass % d | 0.1 | 1.74 |
| Oxygen | Mass % d | 0.1 | 26.1 |
| Net CV | MJ/kg ar | 0.1 | 17.77 |
| Net CV | MJ/kg d | 0.1 | 19.81 |
| Net CV | Kcal/kg ar | 120 | 4244 |
| Net CV | Kcal/kg d | 120 | 4732 |
| Total EF | t $CO_2$/TJ ar | 1 | 93.6 |
| Biomass content by carbon ratio | Mass % d | - | 74 |

**Table 4.** Metals content of RDF.

| Parameter | Unit | LOQ | Result |
|---|---|---|---|
| Arsenic | mg/kg d | 2 | 16 |
| Lead | mg/kg d | 3 | 140 |
| Cadmium | mg/kg d | 0.3 | 3.5 |
| Chromium | mg/kg d | 1 | 330 |
| Copper | mg/kg d | 2 | 520 |
| Nickel | mg/kg d | 1 | 310 |
| Zinc | mg/kg d | 1 | 1700 |
| Mercury | mg/kg d | 0.1 | 2.1 |
| Thallium | mg/kg d | 0.4 | 0.6 |
| Antimony | mg/kg d | 6 | 27 |
| Cobalt | mg/kg d | 1 | 70 |
| Vanadium | mg/kg d | 1 | 200 |
| Potassium | mg/kg d | 50 | 920 |
| Sodium | mg/kg d | 50 | 1100 |
| Phosphorus | mg/kg d | 10 | 1400 |
| Tin | mg/kg d | 10 | 74 |
| Manganese | mg/kg d | 5 | 280 |
| Sum Cd, TI | mg/kg d | - | 4.1 |
| Sum Sb, As, Pb, Cr, Co, Ni, V | mg/kg d | - | 1093 |

**Table 5.** Halides and sulfur content of RDF.

| Parameter | Unit | LOQ | Result |
|---|---|---|---|
| Organic chlorine | Mass % d | 0.05 | 0.2 |
| Bromine total | Mass % d | 0.025 | <0.025 |
| Iodine total | Mass % d | 0.025 | <0.025 |
| Fluorine total | Mass % d | 0.005 | 0.02 |
| Sulfur total | Mass % d | 0.01 | 0.9 |
| Chlorine total | Mass % d | 0.05 | 0.26 |

**Table 6.** Organics content of RDF.

| Parameter | Unit | LOQ | Result |
|---|---|---|---|
| TOC | Mass % d | 0.1 | 49.8 |

In the course of these experiments, silica sand was employed in conjunction with RDF, with the primary objective of reducing the size of the RDF particles through the use of a hammer mill. Silicon sand primarily consists of two main elements: oxygen and silica. More specifically, silica sand is primarily composed of silicon dioxide ($SiO_2$). Quartz, a crystalline mineral constituted by silicon dioxide, is known for its chemical inertness and relatively high hardness, rating 7 out of 10 on the Mohs hardness scale [16]. The selection of this particular type of sand, denoted as ZK, was driven by its finer texture and its absence of unwanted additives, making it a preferable choice over sea sand or sandy clays All the tested materials can be seen in Figure 1.

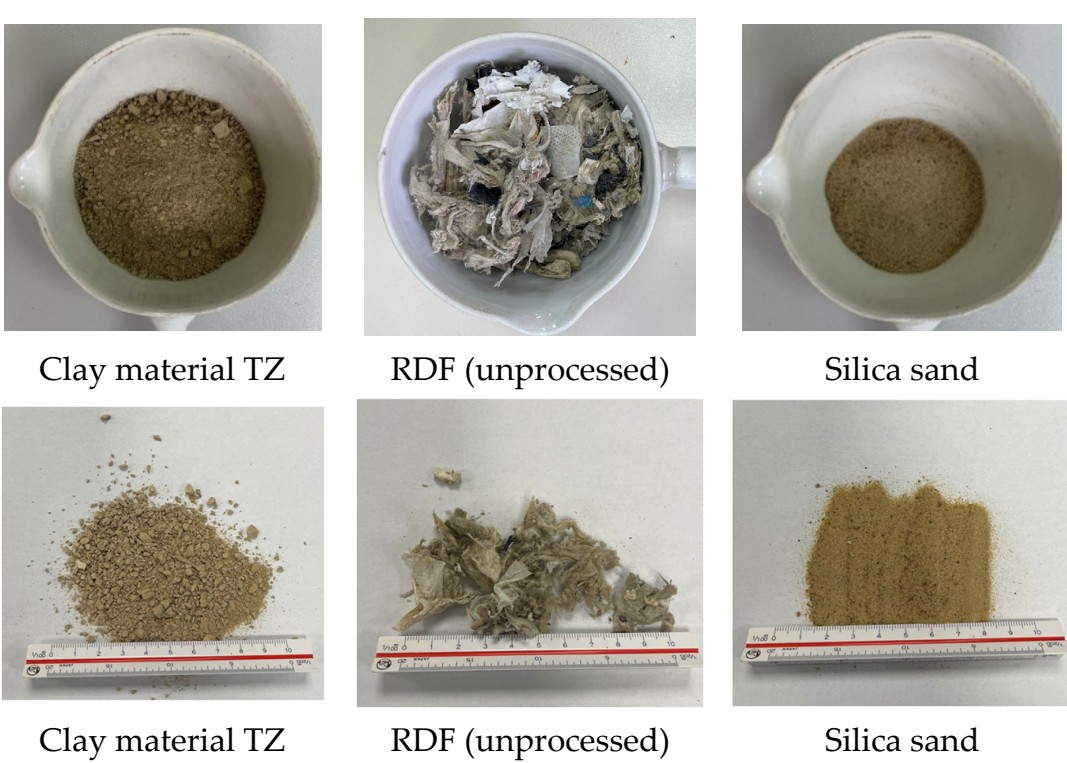

Figure 1. Materials used in the current study.

A significant challenge encountered in the processing of Refuse-Derived Fuel (RDF) arises from its composition, predominantly comprising lightweight plastic and paper materials. These components pose difficulties during the grinding process within the hammer mill, particularly given the elevated temperatures generated inside the mill. Notably, these RDF particles face obstacles passing through metallic screens with a 2 mm mesh size due to their nature. To address this challenge and facilitate the study, a pragmatic solution was implemented. For the processed RDF, a 1:1 mixture of RDF and sand was adopted, effectively streamlining the grinding process for the RDF component. This strategic blending aimed to enhance the grindability of RDF within the hammer mill while maintaining compatibility with the machinery. The grain size of the sand used in this mixture is referenced in detail in Table 7, providing insights into the characteristics of the composite material. Similarly, for the non-processed RDF, the challenges related to grinding were mitigated through a comparable 1:1 mixture with sand, demonstrating a consistent and controlled approach in overcoming the inherent obstacles posed by the lightweight and temperature-sensitive nature of the RDF components. This approach ensured a more effective and uniform grinding process, facilitating subsequent phases of the study on RDF incorporation in ceramic mass for brick production.

**Table 7.** Particle size of the sand grains, fractions 0.063 to 2 mm.

| >2 mm | ASTM 10 | 0.00% |
|---|---|---|
| 0.71 mm | ASTM 25 | 0.15% |
| 0.60 mm | ASTM 30 | 0.15% |
| 0.50 mm | ASTM 35 | 0.90% |
| 0.40 mm | DIN 16-1171 | 4.80% |
| 0.30 mm | ASTM 50 | 19.64% |
| 0.20 mm | DIN 30-1171 | 53.82% |
| 0.10 mm | DIN 60-1171 | 19.80% |
| 0.063 mm | ASTM 230 | 0.65% |

*2.2. Methods*

The initial phase of this study involved the drying of all the aforementioned materials in a laboratory electric dryer of the SCN/400/DG model, maintaining a temperature of 105 °C for a duration of 24 h. Subsequently, the clay and sand materials were subjected to preliminary crushing using a jaw crusher, specifically, model A92, featuring jaws with a 2 mm clearance. In the case of the clay material and the RDF–sand mixture, intended for the formulation of mixture no. 3 (TZRDF10P), these materials underwent further processing through a laboratory hammer mill, designated as Mod. HM/530 Series. This mill utilized a 1 mm mesh screen to facilitate the comminution process.

The materials were meticulously weighed according to their respective mixing ratios, taking into account their prevailing moisture content. Subsequently, these materials were thoroughly blended according to the specific formulation for each mixture. This blending process occurred within a kneading mixer, where the requisite amount of water was carefully introduced. The addition of water was meticulously controlled until the optimal plasticity index was achieved, determined through the Pfefferkorn's test.

The Pfefferkorn plasticity method hinges on the observation of a sample's deformation in response to the calibrated plate's descent onto the underlying test body, shaped with the aid of an auxiliary shaping tool [17]. This test employs two distinct reading scales: one measuring the deformation in millimeters and the other determining the deformation in line with the Pfefferkorn theory. For our study, the Pfefferkorn plasticity tester employed was Ceramic Instruments 01CI4540, and we adopted the calculation method described by Andrade et al. [18]. It's important to highlight that the water added varied for each mixture and depended on how absorbent the clay material was and the extrusion process used for each type of final product.

The uniformly blended mixture for each test underwent extrusion through a vacuum–extrusion process to form rectangular samples of standardized dimensions [19]. The laboratory extruder, specifically the HANDLE KHS-Type: PZVM8b model, was employed for this purpose. The wet material, post-mixing, was loaded into the feeding chamber, equipped with an upper porch for material input, followed by a pre-extruder mixer incorporating a screw mixer responsible for propelling the material through an air vacuum chamber and out of the extruder's outlet. The extrusion process was carefully monitored, including the monitoring of pressure levels through a pressure gauge. The extruder's outer section allowed for the incorporation of interchangeable molds, facilitating the production of extruded products in various sizes and shapes. All the extruded samples were solid, lacking any hollow spaces within their mass, and conformed to a standardized size of $120 \times 20 \times 20$ mm (length × width × height). Notably, the vacuum pressure applied during the extrusion process remained consistent and uniform across all tested mixtures, consistently registering at 0.8 kp/cm$^2$. The plasticity, as assessed by the Pfefferkorn method, fell within the range of 0.7 to 0.9 for all mixtures, achieved by carefully adjusting the water content. In total, each test involved the construction of 15 individual samples. Consequently, across all 3 mixtures examined, a total of 45 samples were meticulously prepared (3 mixtures × 15 samples).

### 2.3. Drying Procedure

The drying process for all extruded specimens was conducted systematically within a laboratory electric oven of type SCN/400/DG. This drying cycle comprised three distinct phases [20], each demanding meticulous attention to address potential issues related to the samples.

The initial "humidity" phase was characterized by maintaining a high ambient humidity within the dryer, an essential step to keep the surface pores of the bricks open. This phase was particularly critical as any mismanagement may have resulted in the development of cracks, deformations, or fragility in the bricks.

Subsequently, during the "shrinkage critical point phase", it was imperative to ensure that the drying shrinkage was completed before the temperature increased significantly for the final drying phase. It was crucial to manage the temperature rise gradually to prevent any cracking issues.

In the last phase, the primary objective was to minimize the remaining body humidity within the bricks as much as possible. All regulations and settings during these phases were tailored to the specific production mixture and its unique behavior.

The initial focus of the drying process was to keep the surface pores of the samples open to facilitate the gradual loss of humidity from the internal body. This phase represented the most critical juncture. In the subsequent phase, as the temperature rose and the dryer humidity decreased, any mishandling could lead to the occurrence of cracks, deformations, or increased fragility in the bricks. The dried samples resulting from this carefully orchestrated drying process are illustrated in Figure 2.

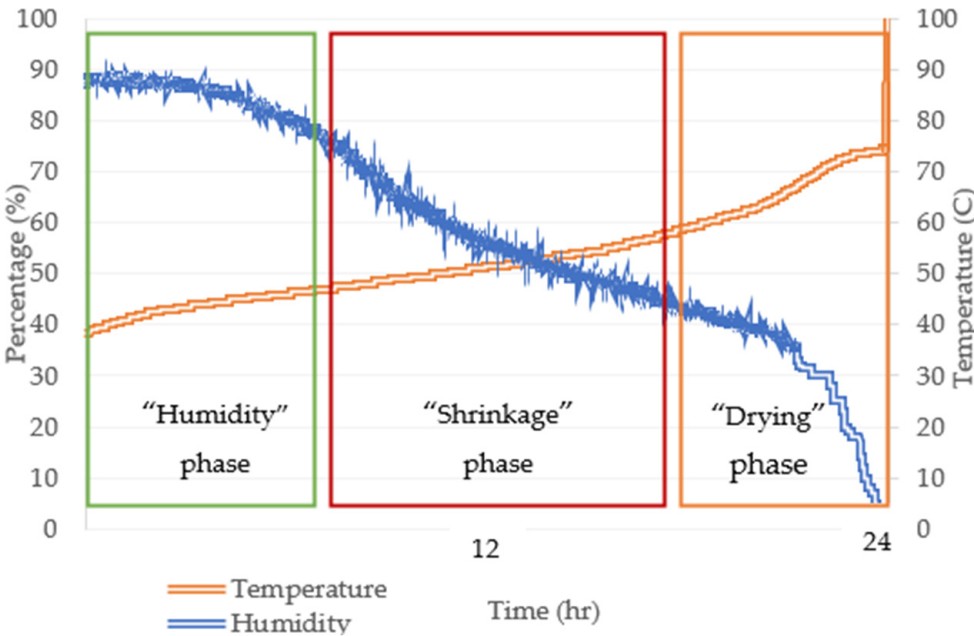

**Figure 2.** Drying circle and each of the 3 phases followed for the tests.

### 2.4. Drying Sensitivity of Samples

The drying sensitivity assessment for the formulated mixtures involved the individual application of Bigot's curve method [21]. Under controlled conditions, the test samples were exposed to a temperature of 25 °C within the laboratory dryer, with the humidity level maintained at a stable 75% inside the dryer. Bigot's curve, a graphical representation, illustrated the fluctuations in water content relative to linear shrinkage over a 24-h period.

Bigot's curve method divides the drying process into two distinct phases. The initial phase, characterized by a constant and rapid drying rate, is indicated by a high linear correlation coefficient. This phase remains consistent until the critical point is reached.

Subsequently, in the second phase, the drying rate gradually decreases, and shrinkage approaches its conclusion [22].

The laboratory dryer, a crucial component of this study, was equipped with the necessary instruments to analyze the drying shrinkage behavior of the green brick samples. Comprising three essential sections—the tunnel dryer unit, the air preparation unit, and the control system incorporating measurement sensors and data acquisition—the dryer had a volume of approximately 125 cm$^3$ and was thoughtfully insulated to minimize heat loss to the surroundings.

Within the air preparation unit, an adjustable centrifugal fan and an adjustable electrical heater played key roles. The centrifugal fan drew drying air from the ambient surroundings. The temperature of the drying air was precisely controlled through a PID-regulated system (Jumo Dtron 304). The drying air, initially passed through an electrical heating zone, then flowed over the sample, maintaining a parallel trajectory to the surface of the sample positioned on a wire mesh. The distance between the hot air inlet and outlet within the tunnel dryer unit was 75 cm, with a 35 cm separation from the air inlet to the sample holder.

To establish steady-state testing conditions, the air fan and electrical heater were initially engaged. Subsequently, a green brick sample was placed on the metallic carrier. Throughout the experiments, the humidity level of the drying air was meticulously monitored. Relative humidity within the dryer was measured every 5 min using a humidity sensor (TMI Orion—CeriDry), adhering to the precise measurement procedures and methodology outlined by Makrygiannis and Karalis [23]. The drying sensitivity level was determined by the CSB index, an indicator of drying sensitivity according to Bigot. The classification of sensitivity levels is presented in Table 8.

**Table 8.** Classification of drying sensitivity according to Bigot's CSB index.

| Classification of CSB | |
|---|---|
| <1.0 | Insensitive |
| 1.0–1.5 | Medium-sensitive |
| 1.5–2.0 | Sensitive |
| >2.0 | Highly sensitive |

*2.5. Measurements*

The determination of weight losses was executed using the Kern FKB 36K0.2 laboratory scale. Linear drying shrinkage and firing linear shrinkage measurements followed the standards outlined in ASTM C326-09. Bending strength evaluations for the dried samples were conducted using test specimens sized at 120 × 20 × 20 mm. A three-point bending test device with a 100 mm distance between the support points was employed for this purpose. Each composition and production method was represented by 3 test specimens.

To calculate the water content after drying, a set of 15 test samples from each preparation procedure, each measuring 120 × 20 × 20 mm, were weighed immediately after shaping. Subsequently, these samples underwent a 24-h drying cycle within the laboratory oven, as previously described. The preparation water content was derived from the wet and dry weight using the formula specified below:

$$\text{WR} = \frac{Weight\ of\ wet - Weight\ of\ dry}{Weight\ of\ dry} \times 100 \tag{1}$$

In order to avoid any misunderstandings, the weight of the dry specimens was used as a reference throughout.

## 3. Results

This study involved a meticulous examination of the results obtained from the constructed mixtures, ensuring uniformity in the sample testing conducted within the same

process environment. A total of fifteen samples were tested, each constructed from mixtures comprising TZ clay and Refuse-Derived Fuel (RDF). The RDF was added to TZ clay at a controlled ratio of 10 wt.-%, adhering to the maximum allowable quantity for lightweight additives in a brick and tile factory. This cautious approach was taken to prevent any potential negative impacts on the final product's strength and quality as exceeding this limit could compromise the structural integrity of the bricks. The mixing process involved the thorough blending of clay and additives for 40 min, utilizing conical rotated mixer model MI/10 to ensure homogeneity in the composite material.

The experimental phase encompassed a comprehensive set of tests spanning the preparation, extrusion, and drying phases. This multifaceted approach aimed to assess variations in both mechanical and physical properties and their correlation with the production environment. To distinguish between the different mixtures, each was systematically labeled with specific abbreviations. For instance, mixtures comprising TZ clay with unprocessed RDF were denoted as TZRDF10, whereas mixtures incorporating processed RDF were labeled as TZRDF10P. This study placed significant emphasis on three critical aspects of the production process, delving into seven key parameters outlined in detail in Table 9. This detailed testing and labeling strategy allowed for a nuanced understanding of the impact of RDF incorporation on the mechanical and physical properties of the produced bricks within the specified production environment.

**Table 9.** Seven key parameters of the production process that emphasis was given to.

| Preparation | Extruding | Drying |
| --- | --- | --- |
| Grinding of the materials for extruding | Extrusion water Plasticity | Sensitivity Shrinkage Bending strength Density |

### 3.1. Preparation

One of the most challenging aspects of this study was the need to break down the RDF (Recycled Demolition Waste) into pieces smaller than 3 mm. This step was crucial to ensure that the RDF could be effectively mixed with the clay material, preventing any issues related to uneven grain size distribution that might affect the homogenization of the mixture with water. It was also essential to avoid potential problems arising from stresses during extrusion and difficulties encountered when cutting the wet body into standard samples measuring 120 mm in length.

In the brick industry, it is not common to employ grinding machines for lightweight materials like RDF. This is primarily due to concerns related to the capacity of such machinery and the associated electricity costs. In this study, an innovative approach was adopted to address this challenge. Instead of using a conventional grinding machine, the research team utilized a standard piece of equipment commonly found in the dry preparation process of brick and tile production known as a hammer mill. The key innovation was mixing the RDF with sand at a 1:1 ratio within the hammer mill, as can be seen in Figure 3.

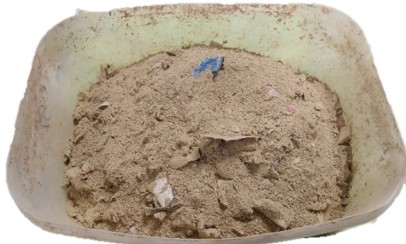 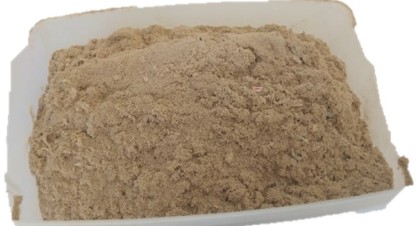

**Figure 3.** (**Left**): mixture with unprocessed RDF (TZRDF10). (**Right**): mixture of processed RDF (TZRDF10P).

This approach proved highly effective for several reasons [24]. Firstly, it eliminated issues related to the melting of the RDF inside the mill as the combination with sand helped maintain a stable processing environment. Secondly, the hardness of the sand prevented excessive wear and tear on the mill's sieves. Importantly, this method yielded exceptional results and ensured that no complications arose during subsequent procedures, such as extrusion and drying.

### 3.2. Extruding

During the extrusion process, it was observed that the mixture containing processed RDF exhibited a noticeable increase in the plasticity of the wet mixture, requiring approximately 1% more extrusion water compared to the 100% clay mixture. Conversely, the mixture containing unprocessed RDF (TZRDF10) displayed better plasticity, requiring nearly the same amount of water as the reference mixture (100% clay—TZ mix) for extrusion. However, challenges emerged during the subsequent cutting phase, where the wet product was shaped into samples [25].

The difficulties encountered during cutting were primarily attributed to the size of the RDF particles. The cutter wire tended to catch and drag the larger RDF pieces, resulting in elevated pressure on the loam material. This, in turn, led to significant stresses, hindered the cutting of specimens, and even caused the detachment of material along the cutting edges. These issues highlighted the unique challenges posed by the presence of unprocessed RDF within the mixture, particularly when it came to the precision of the cutting process (Figure 4). The plasticity and the necessary water for all mixtures can be seen in detail in Figure 5.

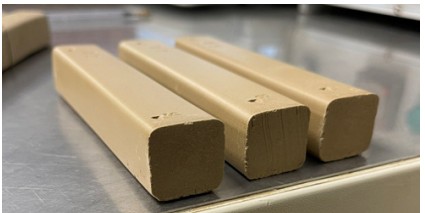 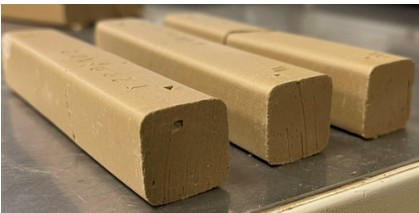 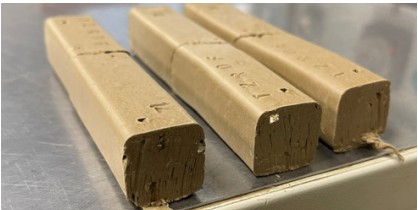

**Figure 4.** (**Left**): 100% clay after cutting. **Middle**: Processed RDF mix after cutting. (**Right**): Unprocessed RDF mix after cutting.

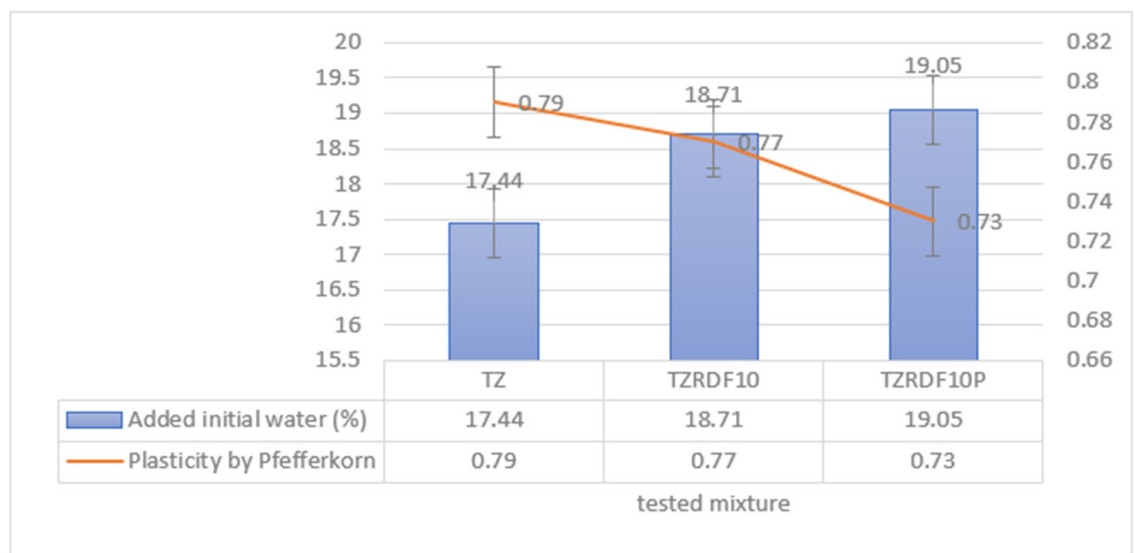

**Figure 5.** Plasticity by Pfefferkorn and necessary extrusion water for all mixtures.

*3.3. Drying*

The drying results of this study revealed a reduction in sensitivity when RDF (Refuse-Derived fuel) was incorporated into the mixtures. The decrease in sensitivity, as shown in Figure 6, was more pronounced in the case of mixtures containing unprocessed RDF, although it was not significantly different compared to the mixtures with processed RDF. This indicates that both types of RDF had a mitigating effect on sensitivity during the drying process.

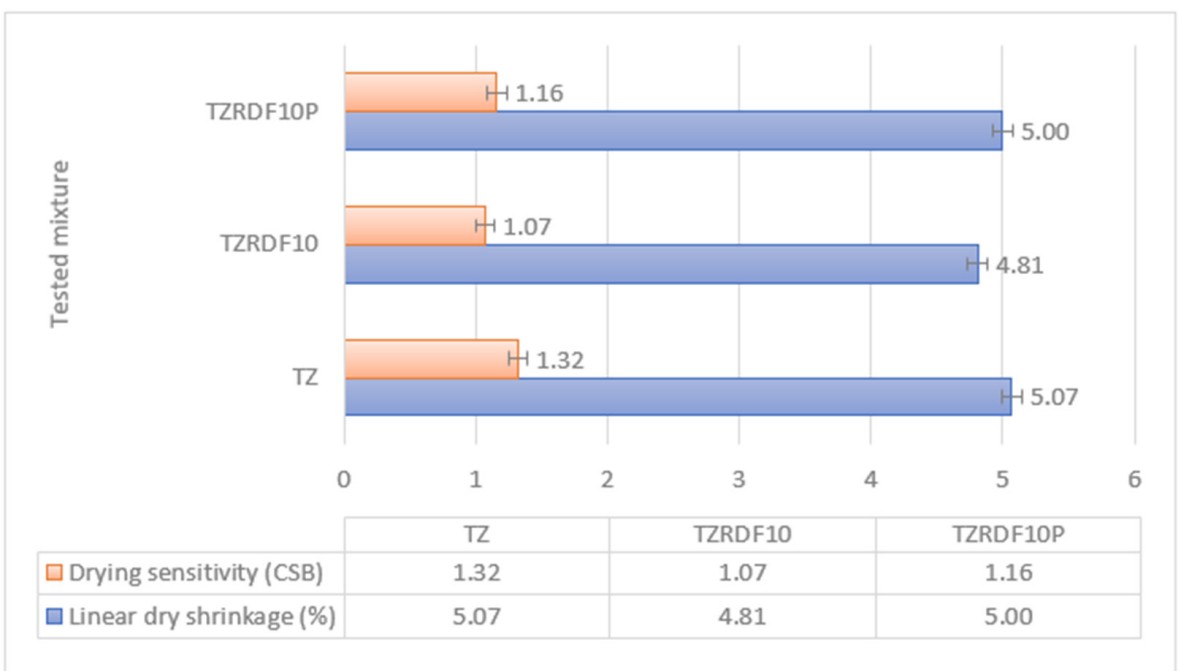

**Figure 6.** Drying linear shrinkage and drying sensitivity of the dried samples.

The drying shrinkage results paralleled the sensitivity findings, with the lowest decrease observed in the mixture containing unprocessed RDF (TZRDF10). This suggests that the unprocessed RDF had a more prominent impact on reducing drying shrinkage, further supporting the overall reduction in sensitivity [26]. Elevated initial water content in clay mixtures leads to increased water evaporation during drying. As water dissipates, clay particles move closer, causing significant volume reduction or shrinkage. This phenomenon underscores the direct correlation between initial water content and drying shrinkage. Essentially, higher initial water content intensifies the structural changes in the clay during the drying process [27].

However, it is noteworthy that the inclusion of RDF had a significant adverse effect on decreasing the bending strength of the dry product, as illustrated in Figure 7. The decrease in bending strength observed in dried clay samples with the addition of RDF can be attributed to several key factors, primarily tied to the alteration of the material's microstructure. The incorporation of RDF, whether processed or unprocessed, introduced a higher level of porosity into the brick body. Porosity refers to the presence of void spaces or pores within the material, and an increase in porosity is commonly associated with a reduction in mechanical strength [28].

In the case of unprocessed RDF-inclusive bricks, the larger size of pores tended to be more prominent compared to those in bricks incorporating processed RDF. Larger pores create weak points within the material, diminishing its overall structural integrity and making it more susceptible to stress and deformation. The lower bending strength in unprocessed RDF-inclusive bricks can be directly linked to these larger, less uniform, void spaces [29].

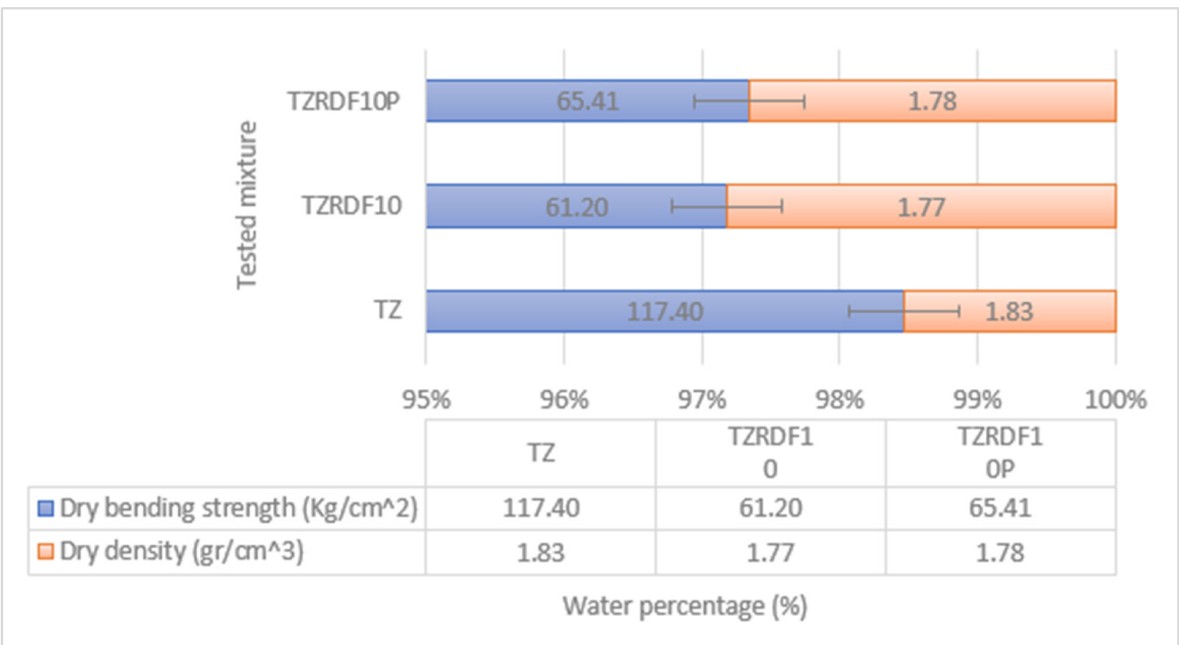

**Figure 7.** Drying bending strength and density of the dried samples.

Moreover, RDF incorporation also resulted in a decrease in the dry product's density. This decrease in density implies that the product became lighter, which can be a crucial factor in a brick and tile factory, especially when considering the overall weight and handling of the final product. The reduction in density, while beneficial in terms of weight, should be balanced with the potential impact on the product's structural strength, as highlighted by the decrease in bending strength.

Furthermore, the incorporation of RDF in the ceramic mixtures exerted a discernible influence on the porosity of the resulting brick samples. Porosity is a critical factor in assessing the performance and functionality of construction materials. The introduction of RDF, particularly unprocessed RDF, led to an increase in porosity within dry products. Elevated porosity implies the presence of void spaces or pores within the material structure, which can impact several key properties, including thermal insulation, permeability, and resistance to external factors.

The investigation into the re-absorption of the dry product shed light on the impact of incorporating Recycled Demolition Waste (RDF) into construction mixtures. After subjecting samples of three different mixtures—the TZ mixture (100% clay), TZRDF10 (10% unprocessed RDF), and TZRDF10P (10% processed RDF)—to complete drying, the subsequent re-absorption of humidity over a 24-h period in room temperature conditions revealed noteworthy findings. The recorded re-absorption percentages of 2.48%, 2.71%, and 2.63% for the TZ, TZRDF10, and TZRDF10P mixtures, respectively, signify an increase in porosity induced by the incorporation of RDF, as can be seen in Figure 8. The elevated re-absorption percentages suggest that the introduction of RDF, whether processed or unprocessed, results in a more porous material structure, allowing for increased moisture uptake during subsequent exposure.

The observed correlation between re-absorption and porosity underscores the intricacies of the material's microstructure and its responsiveness to environmental conditions. Higher porosity, as induced by RDF inclusion, provides more avenues for water absorption, which can have implications for the material's durability and performance over time. Understanding and managing the re-absorption characteristics are crucial in assessing the material's behavior in real-world scenarios, particularly in construction applications where exposure to varying environmental conditions is inevitable.

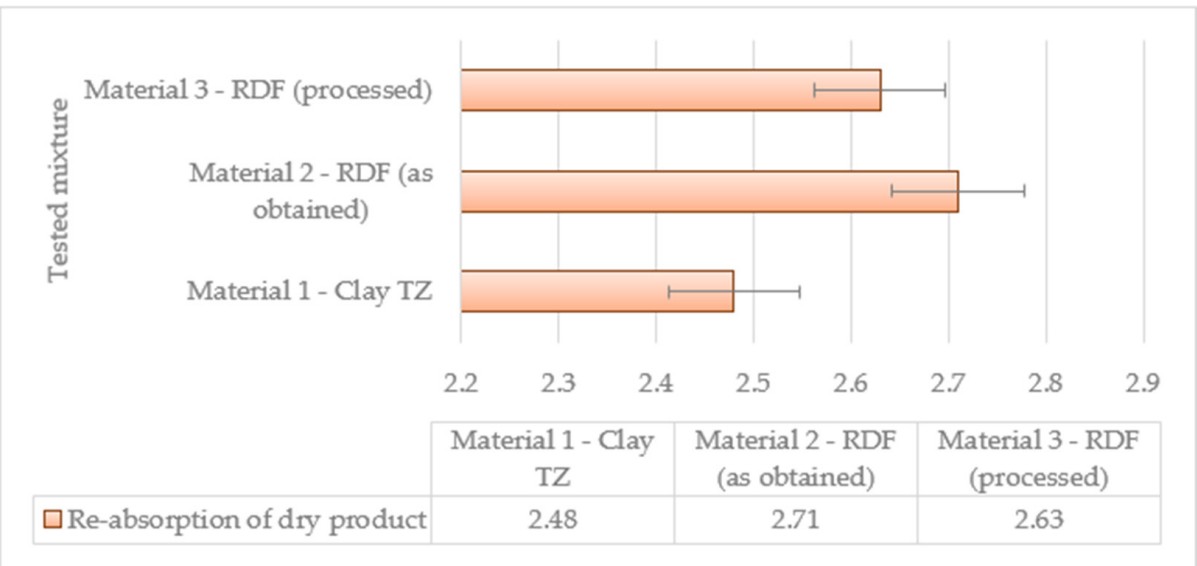

**Figure 8.** Re-absorption of dry products over 24 h in room conditions.

These findings not only contribute to our comprehension of the influence of RDF on porosity but also emphasize the need for a comprehensive understanding of how such changes impact the overall performance and longevity of construction materials [30].

In order to highlight the value of incorporating the tested RDF (Recycled Demolition Waste) into the clay ceramic mass, an extensive compilation of results obtained from the research's process environment was undertaken. These findings have been meticulously documented and are available for review in Table 10.

**Table 10.** Mixture proportions and gathered results.

| Mixture | | TZ | TZRDF10 | TZRDF10P |
|---|---|---|---|---|
| | | wt.-% | wt.-% | wt.-% |
| Material 1—Clay TZ | TC1 | 100 | 90 | 90 |
| Material 2—RDF (as obtained) | TC2 | - | 10 | - |
| Material 3—RDF (processed) | TC3 | - | - | 10 |
| Plasticity by Pfefferkorn | | 0.79 | 0.77 | 0.73 |
| Added initial water | wt.-% | 17.44 | 18.71 | 19.05 |
| Linear dry shrinkage | % | 5.07 | 4.81 | 5.00 |
| Drying sensitivity (Bigot's curve) | CSB | 1.32 | 1.07 | 1.16 |
| Bending strength/dry | Kg/cm$^2$ | 117.40 | 61.20 | 65.41 |
| Re-absorption of dry product | % | 2.48 | 2.71 | 2.63 |
| Remaining water on dry program | % | 1.35 | 1.24 | 1.28 |
| Body density of dry material | gr/cm$^3$ | 1.83 | 1.77 | 1.78 |

Furthermore, for a visual representation of these results, detailed microphotographs can be found in Figure 9. These visual aids provide a more comprehensive understanding of the impact of RDF integration on the clay ceramic mass, shedding light on the microscopic aspects of the material and further enhancing the study's comprehensiveness [31].

In Figure 9, a comparative analysis of different ceramic samples reveals distinct surface characteristics. The surface of the pure clay sample (100% clay material), depicted in Figure 9a, exhibited a notably smoother texture when contrasted with the mixtures incorporating processed and unprocessed Recycled Demolition Waste (RDF) in the ceramic mass. Specifically, Figure 9c illustrates that the mixture with unprocessed RDF resulted in localized inflation, attributed to the presence of coarse RDF particles. This inflation led to an uneven surface, diminishing the overall smoothness of the ceramic material.

Examining the cutting areas of the brick samples in Figure 9d–f, the impact of the cutting process becomes apparent. The wire used for cutting the wet samples into the specified dimensions of 120 × 20 × 20 mm left distinct "scratches" on the surface cutting area. Notably, the surface of the ceramic material containing unprocessed RDF, as depicted in Figure 9f, appeared to be more rugged. This roughness suggests that the wire caused surface damage, resulting in a tactile perception of coarseness in the affected area. Overall, these observations underscore the influence of RDF processing on the surface characteristics of ceramic products.

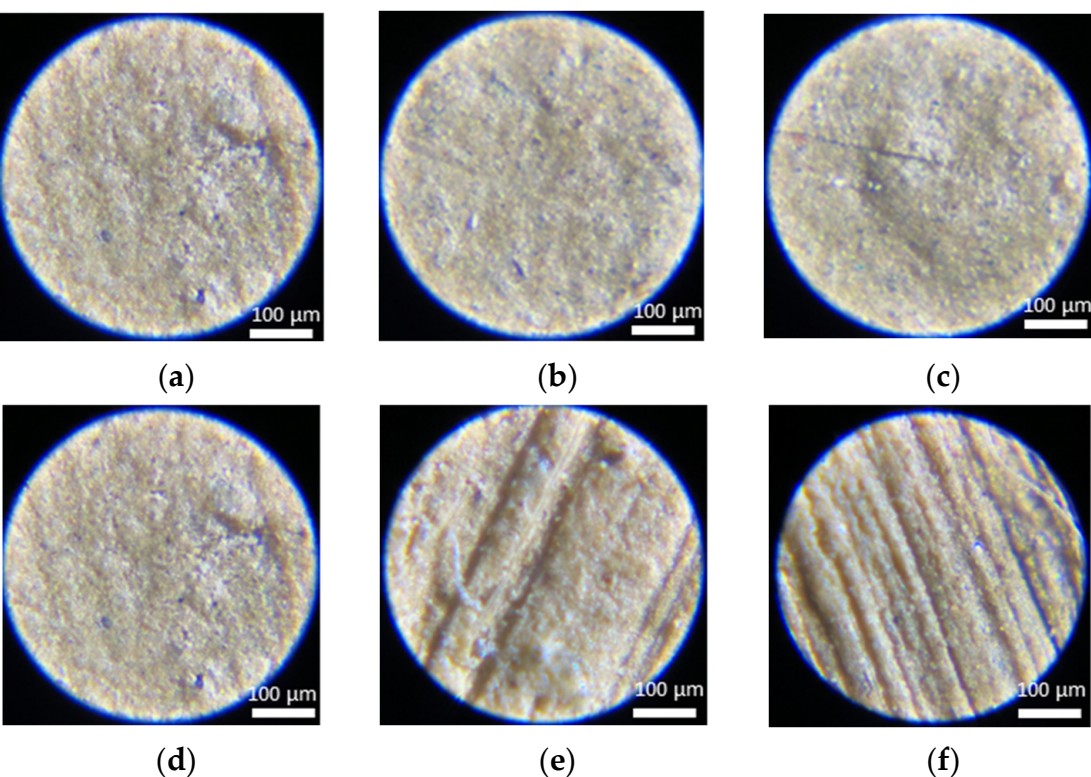

**Figure 9.** (**a**) Surface of dry sample from 100%. (**b**) Surface of dry sample from TZRDF10 (unprocessed RDF). (**c**) Surface of dry sample from TZRDF10P. (**d**) Side view from the cutting area of dry sample from TZ. (**e**) Side view from the cutting area of dry sample from TZRDF10. (**f**) Side view from the cutting area of dry sample from TZRDF10P.

## 4. Discussion

The utilization of RDF (Recycled Demolition Waste) plays a pivotal role in diverting waste from landfills, consequently alleviating the demand for new landfill space. This sustainable approach contributes to environmental protection and promotes responsible waste management practices. Notably, components such as carbides and inorganic elements within RDF undergo combustion at temperatures exceeding 550 °C. This quality prompted our investigation into the feasibility of incorporating RDF as an additive in ceramic mass intended for brick production, where firing temperatures often surpass 850 °C.

Our study is divided into two distinct phases. The first part, which we discussed in this work, focused on the preparation of RDF, its incorporation into ground clay materials, and its impact on extrusion and drying processes. The second part, concerning the firing procedure for RDF as an additive in brick mass, will be the subject of future research. In this current study, we concentrated on the initial phase, where we developed methods for efficiently utilizing RDF as an additive and successfully grinding it without encountering any issues. Furthermore, we delved into how the size of RDF particles influences the quality of vacuum-extruded bricks and their drying properties.

Initially, we crafted two mixtures by incorporating RDF into the clay mass, which were subsequently mixed, extruded, cut into specified dimensions, and dried using a simulated drying program mimicking a real brick plant [32]. We compared the results to those of a 100% clay mixture, which was used as a reference to showcase how RDF and its particle size impact crucial parameters during extrusion and drying. These parameters include the following:

a. The amount of water required for extrusion and the plasticity of the wet mixture.
b. Drying sensitivity, as illustrated by Bigot's curve.
c. Drying linear shrinkage and strength.
d. The density of the dry samples and their reabsorption from the environment.

The results of our study indicate that our innovative process for RDF implementation, involving grinding methods and mixing with clay materials using brick and tile industry machinery, yielded promising results for the production mixture of RDF and clay materials. Incorporating sand, as explained in the results section, facilitated the hammer mill's ability to grind materials to particle sizes smaller than 2 mm. This enhanced the mixing process, stabilizing it and reducing operational costs by safeguarding the machine from wear and tear caused by the hard grains of sand [33]. Importantly, this process did not generate excessive heat within the hammer mill's grinding chamber due to the presence of sand.

Extrusion tests revealed that the mixture containing unprocessed RDF material posed challenges during the extrusion and brick-cutting processes. The large RDF particles were difficult to cut efficiently with the wire cutter, resulting in numerous specimens exhibiting cracks, stress, and unstable dimensions. In contrast, using ground RDF demonstrated considerable improvements, reducing these issues by nearly 95%. Additionally, the inclusion of RDF enhanced the plasticity of the mixture, requiring less extrusion water. This not only conserved water resources but also reduced energy consumption during the drying of samples, translating to cost savings for brick plant operations [34].

In terms of drying behavior, the samples displayed reduced sensitivity, as indicated by Bigot's curve, which correlated with decreased drying linear shrinkage. The RDF acted as an inert additive within the ceramic mass [35]. Another significant finding from this study was the 5% reduction in the density of the dried products in the RDF-containing mixtures. This suggests that, following the firing procedure, the target of achieving a 10% reduction in density is attainable. A lower density implies increased porosity in the mass, which contributes to improved thermal insulation properties for the final products.

The presence of metals within the composition of Recycled Demolition Waste (RDF) poses significant challenges in the context of brick production. When RDF contains a higher metal content, the extrusion machinery becomes susceptible to increased wear and tear, presenting a considerable risk of accelerated deterioration. Metals, inherently abrasive in nature, can lead to heightened abrasion on equipment surfaces, potentially causing damage that necessitates frequent maintenance and, consequently, operational downtime. Beyond the mechanical implications, the influence of metals on porosity and mechanical strength, as previously discussed in the RDF context, assumes added complexity [36]. Metal particles within the mixture have the potential to introduce additional pores and disrupt the microstructure, thereby intricately affecting the overall behavior of the brick during the drying phase. As a result, the meticulous removal of metals from RDF compositions emerges as a crucial prerequisite to mitigate these adverse effects, ensuring the longevity of machinery, minimizing maintenance-related challenges, and preserving the structural integrity and performance characteristics of the final ceramic products [37].

## 5. Conclusions

In summary, our study highlights the potential of incorporating Recycled Demolition Waste (RDF) as an additive in ceramic mass for use in the brick industry, including for products like thermoblocks. This not only contributes to sustainable waste management but also protects the environment by reducing the reliance on landfills. The combustion properties of RDF at temperatures above 550 °C make it a valuable candidate for enhancing

clay-based materials, especially in the brick production process, where firing temperatures often exceed 850 °C.

Our research comprised two phases, with the first part focusing on the preparation of RDF, its integration with clay materials, and its impact on the extrusion and drying phases. Our innovative approach, utilizing brick and tile industry machinery and incorporating sand, yielded promising results. Grounding RDF particles to less than 1 mm facilitated the mixing process, reduced operational costs, and protected machinery. This approach also helped maintain stable grinding temperatures within the hammer mill.

During extrusion, the presence of unprocessed RDF material posed challenges, impacting cutting and specimen quality. However, the use of ground RDF significantly mitigated these issues, resulting in a more efficient and cost-effective process with improved plasticity and reduced water requirements. These findings have practical implications for brick plant operations and can lead to resource and energy savings.

The drying behavior of the samples further supported the benefits of RDF integration, showing reduced sensitivity, decreased drying linear shrinkage, and improved density properties. RDF acted as an inert additive, contributing to a 5% reduction in density, leading to increased porosity and enhanced thermal insulation properties, which are particularly important for applications like thermoblocks.

In the brick industry, where durability, thermal performance, and cost-efficiency are crucial, our findings emphasize the potential benefits of incorporating RDF into clay-based materials. While further research is needed to address the firing procedure of RDF as an additive in brick mass, the results from this initial phase underscore the promise of this approach for sustainable and environmentally responsible brick production. By harnessing the advantages of RDF, the brick industry can create products that meet both performance and sustainability goals, paving the way for a more efficient and eco-friendly future.

This study aims to illuminate the potential advantages and challenges associated with incorporating RDF into clay-based products, contributing to sustainability and waste reduction in construction and manufacturing. Our findings offer valuable insights into the performance and feasibility of these mixtures, providing crucial information for industries striving to adopt eco-conscious production methods.

This article not only revealed the applied methodology and experimental setup but also presented results related to the behavior of RDF-inclusive clay block mixtures in the production environment. We anticipate that this research will exert a considerable influence on future practices and policies, contributing to the growing body of knowledge regarding eco-friendly and sustainable manufacturing processes.

**Author Contributions:** Conceptualization, O.P. and K.K.; methodology, A.T. and I.M.; validation, I.M., A.T. and K.K.; formal analysis, I.M. and K.K.; investigation, O.P.; resources, O.P. and K.K.; data curation, A.T. and I.M.; writing—original draft preparation, I.M.; writing—review and editing, A.T., I.M. and K.K.; supervision, A.T. All authors have read and agreed to the published version of the manuscript.

**Funding:** This research received no external funding.

**Data Availability Statement:** The data presented in this study are available upon request from the corresponding author.

**Acknowledgments:** This study was performed as a part of RES.U.REC.T project no. T2EΔK-03668. The authors are grateful to SABO S.A. staff for providing details for a brick and tile industry operation. Special acknowledgment to the XALKIS S.A. company for providing the clay material with the code TZ.

**Conflicts of Interest:** The authors declare no conflict of interest. The XALKIS S.A. company was not involved in the study design, collection, analysis, interpretation of data, the writing of this article or the decision to submit it for publication.

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
