# Peer review of "Assessing the Effects of Refuse-Derived Fuel (RDF) Incorporation on the Extrusion and Drying Behavior of Brick Mixtures"

_ceramics, doi:10.3390/ceramics6040145_

Round 1

Reviewer 1 Report

Comments and Suggestions for Authors

Authors discussed the effect of refuse derived fuel as a waste additive on the extrusion and drying of clay bricks. The methodology of this manuscript seems to be a little simple, and the empirical data is insufficient to fulfil the publication standard of Ceramics. The reviewer would suggest a major revision, and for details please see below:

Line 47 Is there really so many bricks produced annually? 1,600 billion is so unbelievable so the reviewer suggests a double-check.

Lines 54-55 Please note that this manuscript has not involved any test on the thermal insulation of clay blocks, even the porosity of clay bricks has not been determined. So please check throughout the manuscript to delete descriptions about “thermal insulation”.

Lines 76-77 “The second mixture blends nonhazardous RDF with clay in varying 10% to 90% volume ratios”

For the second mixture, the authors only made one sample: 10% RDF (uncrushed) + 90% clay. Why did they declare that RDF with varying volume rations from 10% to 90% was made? This is not the truth.

The literature review in the introduction section is inadequate. The authors should summarize the research progress regarding RDF -inclusive ceramic blocks, and what the research gap is. Currently, the reviewer does not see many similar research mentioned/discussed in the Introduction section.

Lines 82-96 These three paragraphs should be deleted or moved to the Conclusion section.

Line 99-107 These contents belong to Materials and Methods. It is not proper to put them in Introduction.

Lines 120-126 Please delete this paragraph or moved to the Conclusion section.

Lines 129-136 Please delete these contents. Just cut straight to the point.

Line 141 Please specific the amount of water for every mixture.

Line 156 In table 3, what is “Plasticity”? Do you mean Plasticity index?

Line 197 It is silicon instead of silica.

Line 394 According to the text, the middle image in Figure 7 should be processed PDF while the right image refers to the unprocessed one.

Lines 403-404 The reviewer thinks that the variation of drying shrinkage is attributed to different initial water content in these three mixtures. This parameter must be considered when analysing the drying shrinkage data.

Figure 10 Authors only listed microscopic images but didn’t provide any description in the text. Please 1) detail the typical characteristic of these images, 2) add the scale bar in Figure 10, and 3) discuss how RDF affect the microstructure of ceramic blocks.

If 15 samples were run for every mixture, the error bar must be supplied in Figures 6, 8, and 9.

Porosity is one of the most vital properties of dried brick. The effect of RDF on brick porosity should be provided.

Line 420 Figure 9 The addition of RDF decreases the bending strength of dried clay blocks but the underlying reason has not been discussed by authors. Authors should add discussions on this aspect as the mechanical strength is quite important. From the reviewer’s point of view, the lower strength is attributed to higher porosity and of brick body after RDF was incorporated. Unprocessed RDF-inclusive brick shows a lower strength than processed RDF-inclusive brick because larger size of pores tend to exist in the former. For the relationship between pores/cracks and mechanical strength of bricks, the following references are recommended: https://doi.org/10.1016/j.conbuildmat.2020.119346 ; https://doi.org/10.1016/j.conbuildmat.2021.125828

Lines 442-444 “Our study is divided into two distinct phases. The first part, which we discuss in this work, focuses on the preparation of RDF, its incorporation into ground clay materials, and its impact on the extrusion and drying processes”

Authors are allowed to spilt their work into different parts but the reviewer doesn’t think the current data and related discussions can be integrated to a publication fulfilling the discipline standard. More characterisations should be provided, including porosity change of dried ceramic blocks, and interface image between RDF and clay particles employing SEM.

Reviewer 2 Report

Comments and Suggestions for Authors What influence of  metals content of RDF on the  extrusion and drying behavior of brick mixture sholud be pointed out;  

Reviewer 3 Report

Comments and Suggestions for Authors

The paper presents a comprehensive investigation into the effects of incorporating Refuse Derived Fuel (RDF) on the extrusion and drying behavior of brick mixtures. The study aims to address challenges in brick production by exploring solutions through RDF incorporation. However, several revisions are necessary before considering it for publication.

Abstract: The abstract lacks clear contextualization, fails to identify gaps in the literature (GAP), and inadequately explains why the study is conducted. Additionally, the inclusion of numerical data is crucial for a more precise and concise overview of the content.

General Structure: A global restructuring of the text is needed, as it currently resembles a thesis or dissertation, making it lengthy and tiresome. I recommend a more direct and concise approach typical of scientific articles. Figures 1, 3, 5, and 7 are deemed unnecessary and should be removed. The remaining figures require significant design improvements to meet scientific article standards. Visual presentation should ensure clarity and relevance, with any figure not directly contributing to the understanding of results or conclusions to be eliminated.

Tables: There is an excess of tables, and improvements in the organization of the data are crucial. The use of tables should be restricted to situations where graphs are not feasible. A detailed review of the necessity of each table is recommended, eliminating redundancies and ensuring a more efficient presentation of data.

Results: The absence of comparisons with the literature in the results section is a critical gap. The validity of the work depends on contextualizing it with previous research. A more in-depth analysis is recommended, highlighting the influence of thermal treatment temperatures and including data on the mechanical resistance of the obtained products.

Conclusions: More comparisons with the literature should be incorporated into the conclusions, providing a more comprehensive evaluation of the work and helping determine its significant contribution to the field. Ensure that the conclusions align with the objectives and results presented.

Comments on the Quality of English Language

Must be improved

Reviewer 4 Report

Comments and Suggestions for Authors

The subject of using Refuse Derived Fuel (RDF) for the production of ceramics discussed in the article can only be considered as a laboratory case study. In practice, RDF currently available on the market cannot be considered for potential use in the production of ceramics. The physicochemical properties of RDF, due to the heterogeneous nature of the waste used for their production, are very variable, which makes their use impossible. I am not sure whether the authors know the definition of RDF fuel, because in their work (on page 11 line 338) they also use the term "Recycled Demolition Waste" for the abbreviation RDF, and these are completely different waste . For demolition waste (from construction facilities) there are chances of effective use other than incineration plants with energy recovery (RFD is prepared for this purpose).
The article in its current form is not suitable for publication.

The paper lacks a critical analysis of the limitations and implications of the results, especially in the context of structural strength and practical applications. When considering topic, however, as a case study, first of all, a complete analysis of the RDF composition should be provided. What the Authors provide in Tables 6 and 7 - these are parameters required for alternative fuels, not for the material for producing ceramics. Provide its chemical composition and was it uniform? Describe how the quality and consistency of RDF samples was controlled during experiments? Moreover, the authors should state what were the criteria for selecting RDF? It would be worth providing information about the moisture content in the fuel and its control processes.
 The text distinguishes between raw and processed RDF, with processed RDF being milled and shredded before integration. However, the specificity of the processing method, including the degree of grinding and fragmentation, is not clearly defined. The lack of clarity about the parameters, equipment used, and experimental conditions makes it difficult for readers to assess the robustness of the research and process control.

 The results need to be analyzed statistically. Including statistical tests would increase the validity of the study by quantifying the level of confidence in the reported results.

The description of the clay used (Tables 3, 4, 5, 11 and Fig 5) has already been published in the work: Recycling 2022, 7, 75. https://doi.org/10.3390/recycling7050075, so it should be referenced in the text of the work.
The composition given in table 4 is not 100%. Figure 2 with and without a scale (measuring tape) makes no sense. There is no time scale in Fig. 4. Tables 1 and 2 should be provided in the methodology, not in the preliminary part of the work. The diagram in Figure 1 is not a diagram of the procedures used in the ceramics industry. Even for a simplified diagram, it is not clear and correct. The conclusions are not adequate to the subject of the work, and based on the presented results, it is not possible to conclude about the suitability of RDF fuel for the ceramic industry.

Round 2

Reviewer 1 Report

Comments and Suggestions for Authors

The authors have diligently revised the manuscript in line with the reviewer's comments. Consequently, the reviewer now recommends approving the publication of this manuscript in Ceramics.

Reviewer 3 Report

Comments and Suggestions for Authors

Moderate editing of English language required

Comments on the Quality of English Language

Moderate editing of English language required

Reviewer 4 Report

Comments and Suggestions for Authors

The topic of the article is original, although without practical application. The article has been corrected in accordance with the re-reviewer's comments, and may be published in this form.